# Endoscopic Ultrasound-Guided Gastroenterostomy versus Enteral Stenting for Malignant Gastric Outlet Obstruction: A Retrospective Propensity Score-Matched Study

**DOI:** 10.3390/cancers16040724

**Published:** 2024-02-08

**Authors:** Maria Cristina Conti Bellocchi, Enrico Gasparini, Serena Stigliano, Daryl Ramai, Laura Bernardoni, Francesco Maria Di Matteo, Antonio Facciorusso, Luca Frulloni, Stefano Francesco Crinò

**Affiliations:** 1Diagnostic and Interventional Endoscopy of Pancreas, The Pancreas Institute, G.B. Rossi University Hospital, 37134 Verona, Italy; engasparini.eg@gmail.com (E.G.); laura.bernardoni@aovr.veneto.it (L.B.); luca.frulloni@univr.it (L.F.); stefanofrancesco.crino@aovr.veneto.it (S.F.C.); 2Unit of Gastroenterology and Digestive Endoscopy, Campus Bio Medico University, 00128 Rome, Italy; s.stigliano@policlinicocampus.it (S.S.); f.dimatteo@policlinicocampus.it (F.M.D.M.); 3Gastroenterology and Hepatology, University of Utah Hospital, Salt Lake City, UT 84112, USA; daryl.ramai@hsc.utah.edu; 4Gastroenterology Unit, Department of Medical Sciences, University of Foggia, 00161 Foggia, Italy; antonio.facciorusso@unifg.it

**Keywords:** interventional endoscopy, duodenal stenting, self-expanding metal stents, pancreatic cancer, gastric cancer, LAMS

## Abstract

**Simple Summary:**

This study was conducted to compare two endoscopic procedures for managing malignant gastric outlet obstruction. The first and most commonly used procedure is enteral stenting, which is safe, easy, and widely available. The second procedure, endoscopic ultrasound-guided gastroenterostomy (EUS-GE), is more challenging, can result in severe adverse events, and requires specific skills. However, EUS-GE, which bypasses the tumor, could have some advantages in terms of the stent dysfunction rate. Our study found that stent dysfunction occurred more frequently after enteral stenting, while stent patency was longer after EUS-GE. However, the early clinical efficacy was comparable between the two procedures and was influenced by patients’ performance status. This study suggests that the choice between these two techniques should be personalized based on several factors, including patients’ prognosis and life expectancy, clinical success, and local expertise.

**Abstract:**

Background: Endoscopic ultrasound-guided gastroenterostomy (EUS-GE) using lumen apposing metal stent has emerged as a minimally invasive treatment for the management of malignant gastric outlet obstruction (mGOO). We aimed to compare EUS-GE with enteral stenting (ES) for the treatment of mGOO. Methods: Patients who underwent EUS-GE or ES for mGOO between June 2017 and June 2023 at two Italian centers were retrospectively identified. The primary outcome was stent dysfunction. Secondary outcomes included technical success, clinical failure, safety, and hospital length of stay. A propensity score-matching analysis was performed using multiple covariates. Results: Overall, 198 patients were included (66 EUS-GE and 132 ES). The stent dysfunction rate was 3.1% and 16.9% following EUS-GE and ES, respectively (*p* = 0.004). Using propensity score-matching, 45 patients were allocated to each group. The technical success rate was 100% for both groups. Stent dysfunction was higher in the ES group compared with the EUS-GE group (20% versus 4.4%, respectively; *p* = 0.022) without differences in clinical efficacy (*p* = 0.266) and safety (*p* = 0.085). A significantly shorter hospital stay was associated with EUS-GE compared with ES (7.5 ± 4.9 days vs. 12.5 ± 13.0 days, respectively; *p* = 0.018). Kaplan–Meier analyses confirmed a higher stent dysfunction-free survival rate after EUS-GE compared with ES (log-rank test; *p* = 0.05). Conclusion: EUS-GE offers lower rates of stent dysfunction, longer stent patency, and shorter hospital stay compared with ES.

## 1. Introduction

Malignant gastric outlet obstruction (mGOO) is a condition impacting the quality of life in patients with gastrointestinal neoplasms. Traditionally, mGOO was treated using a surgical approach or by endoscopic placement of self-expanding metal stents (SEMS) [1]. Surgical gastrojejunostomy creates a large anastomosis with high rates of technical success with low risk of reintervention; however, it is invasive and can be associated with high rates of morbidity [2]. Compared with surgery, enteral stenting (ES) is associated with lower rates of adverse events (AEs), quicker resumption of oral intake, and shorter hospital length of stay [3]. Nevertheless, a consistent rate of reintervention has been reported after ES due to stent dysfunction following tumor ingrowth or migration [4].

In the last few years, endoscopic ultrasound-guided gastroenterostomy (EUS-GE) with the creation of a digestive anastomosis using lumen apposing metal stents (LAMS) has been increasingly used for the treatment of mGOO [5]. With the availability of large-diameter LAMS [6,7], EUS-GE could combine the advantages of surgical gastroenterostomy (i.e., low rate of stent dysfunction from bypassing the primary tumor) with the less invasivity of ES. Preliminary studies on EUS-GE have reported encouraging results [8,9,10]; thus, EUS-GE has been included in recent ESGE guidelines as an alternative to ES or surgery in an expert setting [11]. In a retrospective study comparing EUS-GE versus ES and laparoscopic gastrojejunostomy, the reintervention rate for recurrent obstruction was lower in the EUS-GE group [12]. Moreover, in a meta-analysis comparing these three procedures, EUS-GE had a higher rate of clinical success with a lower rate of reintervention compared with ES, and a lower rate of severe AEs with comparable clinical success compared with surgical gastrojejunostomy [13].

However, EUS-GE is technically demanding and requires a high level of skill and expertise in interventional EUS [14,15]. Moreover, severe AEs requiring surgery can occur in cases of LAMS misdeployment [16]. For these reasons, EUS-GE is not performed in all centers. On the other hand, ES is safe and effective (particularly for short-term outcomes), widely available, technically affordable, and does not require advanced technical skills in interventional EUS [17]. Therefore, when dealing with mGOO, endoscopists must determine which procedure to employ. However, only a few comparative studies have been published to date [9,18,19,20,21,22].

In this study, we aimed to compare EUS-GE with ES in a large cohort of patients before and after propensity score-matching. The primary aim was the rate of stent dysfunction. Secondary aims were technical success, clinical efficacy, safety, and hospital length of stay.

## 2. Materials and Methods

### 2.1. Study Design and Population

This is a retrospective observational study involving two referral (more than 100 interventional EUS procedures/year) Italian centers (the Endoscopy Unit of University Hospital of Verona and the Endoscopy Unit of Fondazione Policlinico Universitario Campus Bio-Medico of Rome) comparing EUS-GE and ES in mGOO. 

All consecutive adult patients referred for endoscopic treatment (EUS-GE or ES) of mGOO between June 2017 and June 2023 were retrospectively evaluated. Demographical data, pre-procedural clinical data, and technical details of endoscopic procedures were collected. Exclusion criteria were: (1) benign GOO; (2) extensive gastric involvement; (3) multiple strictures; (4) Gastric Outlet Obstruction Scoring System (GOOSS) ≥ 2. 

### 2.2. Definitions

The GOOSS, a scale system ranging from 0 (no oral intake) to 3 (full diet) based on the highest intake tolerability [23] (Table 1), was retrieved from clinical charts and was used to assess the severity of obstructive symptoms before and after the procedure.

AEs were defined according to international lexicon [24] and divided into “procedure-related” and “not-procedure-related”. AE severity was classified according to the AGREE [25] classification and timing was classified as intraprocedural, early (within 7 days), and late (after 7 days). 

The site of stenosis was classified according to the Mutignani classification [26], which considers the anatomical location of the obstruction in relation of the papilla: type I, involving the pylorus/duodenal bulb/upper duodenal genu; type II, involving the second part of the duodenum; type III, involving the third part of the duodenum.

Performance status was evaluated according to the Eastern Cooperative Oncology Group (PS-ECOG) score: 0 = Fully active, able to carry on all pre-disease performance without restriction; 1 = Restricted in physically strenuous activity but ambulatory and able to carry out work of a light or sedentary nature, e.g., light housework, office work; 2 = Ambulatory and capable of all self-care but unable to carry out any work activities; up and about more than 50% of waking hours; 3 = Capable of only limited self-care; confined to bed or chair more than 50% of waking hours; 4 = Completely disabled; cannot carry out any self-care; totally confined to bed or chair; 5 = Dead.

### 2.3. Study Outcomes

The primary outcome was the rate of stent dysfunction, which was defined as the need for any reintervention for GOOSS ≤ 1 after initial clinical success. 

Secondary outcomes included: (1) technical success, defined as the correct placement of LAMS for EUS-GE (evidence of blue-stained fluid flowing into the stomach) or SEMS for ES (covering the full length of the stenosis); (2) clinical efficacy (GOOSS of ≥2 reached within one week); (3) safety, defined as the overall rate of AEs; (4) hospital length of stay (calculated as days from the procedure to discharge).

Moreover, in the propensity-matched population, stent dysfunction-free survival (measured as days between stent placement and the occurrence of stent dysfunction or surgery or death) in the two groups was compared. 

Follow-up data, including AEs and clinical outcomes (chemotherapy, surgery, endoscopic reintervention, death), were obtained from medical records and by telephone interview.

### 2.4. EUS-GE and ES Procedures

EUS-GE were performed by endosonographers with more than 5 years of experience in interventional EUS-guided procedures. The procedures were performed in a suite equipped with fluoroscopy with patients under general anesthesia and orotracheal intubation, in a supine position, and using CO2 insufflation. In all cases, the antegrade direct free-hand technique was used for EUS-GE [5]. 

Firstly, a 3.7 mm working channel gastroscope (Olympus GIFH 1TH190, Tokyo, Japan) was used to place a 0.035-inch guidewire and a 7 Fr catheter/nasobiliary drainage beyond the stenosis. Medium contrast was injected to confirm the correct position of the catheter and reveal the anatomy of the proximal small bowel loops (Figure 1).

The gastroscope was exchanged with a linear array echoendoscope (EG3870UTK or EG38J10UT, Pentax Medical, Tokyo, Japan) and the catheter was connected to a water pump. After identification of the target limb, glucagon or hyoscine butylbromide was administrated intravenously to reduce peristalsis, and the small bowel lumen was irrigated and distended with saline mixed with methylene blue at a flow rate of 500 mL/min. Preferably, the limb closest to the ligament of Treitz was chosen to guarantee more stability. An electrocautery-enhanced LAMS of 20 mm lumen diameter (Hot-Axios™, Boston Scientific, Marlborough, MA, USA) was used to create the anastomosis under both fluoroscopy and EUS guidance (Figure 2).

ES procedures were performed by expert endoscopists with patients under conscious or deep sedation in a supine position. An operative gastroscope, colonoscope, or duodenoscope was employed according to the site of the stenosis. After passing a 0.035-inch guidewire through the stenosis and estimating the stenosis length through the injection of contrast medium, a nitinol uncovered SEMS (Enteral Wallflex™, Boston Scientific, Marlborough, MA, USA) was carefully advanced over the wire under fluoroscopy and deployed across the stenosis. All SEMS were 22 mm wide, while SEMS length (6, 9, or 12 cm) was chosen according to the extent and morphology of the stenosis. At the end of the procedure, the position and the efficacy of the SEMS were systematically checked by contrast injection under fluoroscopy, demonstrating the rapid flow of contrast distal to the stricture (Figure 3). Complete stent expansion was usually checked by abdominal X-ray 24–48 h after the procedure.

### 2.5. Statistical Analysis

The results were summarized by descriptive statistics (mean ± standard deviation (SD) or median with interquartile range (IQR) for continuous variables and frequency distributions for categorical variables). The χ^2^ or Fisher’s exact tests (where appropriate) and the Wilcoxon rank-sum (Mann–Whitney) test were used to compare categorical and continuous variables, respectively. Multivariate analysis was carried out employing stepwise logistic regression to analyze factors impacting primary and secondary outcomes in the overall population. 

To minimize selection bias of the observed data, 1:1 ratio propensity score-matching was conducted based on covariates that are known to affect patient outcomes, including sex, age, PS-ECOG, pre-procedural GOOSS, type of malignancy, biliary obstruction, and presence of ascites. A stringent maximum propensity score difference of 0.05 was used for matching [27].

Survival analysis and patency evaluation were performed using the Kaplan–Meier method and the differences between groups were compared through a log-rank test. All analyses were two-tailed, with *p* < 0.05 considered to be statistically significant, and were performed using SPSS 22 software (SPSS, an IBM company, Chicago, IL, USA).

## 3. Results

### 3.1. Main Study Population

Overall, 212 patients were referred for the treatment of mGOO. Three patients had a benign condition and eleven patients had a GOOSS of 2 which were excluded (Figure 4). A total of 66 patients in the EUS-GE group (38 males, mean age 69.6 ± 11.4 years) and 132 in the ES cohort (68 males, mean age 66 ± 11.9 years) were included.

The most common etiology of mGOO was pancreatic cancer (n = 139, 70.2%) and the stenosis was most frequently type II (n = 82, 41.4%). Overall, previous or immediately subsequent biliary drainage was performed in 128 (64.6%) patients and ascites were present in 34 patients (17.1%). 

The baseline characteristics of the study population are summarized in Table 2. The EUS-GE and ES groups significantly differed in age (*p* = 0.045), PS-ECOG (*p* = 0.005), pre-procedural GOOSS (*p* < 0.0001), type of malignancy (*p* = 0.01), ascites (*p* < 0.001), and biliary obstruction (*p* < 0.001).

#### 3.1.1. Stent Dysfunction

Overall, 195 patients (65 EUS-GE and 130 ES) completed the procedure and were analyzed. Stent dysfunction requiring reintervention was observed in 24 patients (12.3%): 2 in the EUS-GE group (3.1%) and 22 in the ES group (16.9%) (*p* = 0.004). 

In the EUS-GE group, the two stent dysfunction cases were treated with a coaxial enteral SEMS and surgically for concomitant perforation close to the LAMS, respectively. 

In the ES group, the reintervention was surgical (laparoscopic gastroenterostomy) in 1 (4.3%) case and endoscopic in 22 (96.7%) cases, consisting of coaxial SEMS placement (n = 21) and performing an EUS-GE (n = 1). 

In the multivariate analysis, ES (OR 8.55 [95% CI, 1.88–38.82; *p* = 0.005]) and type III stenosis (OR 3.49 [95%CI, 1.35–9.02; *p* = 0.009]) were independent factors associated with stent dysfunction (Table 3).

#### 3.1.2. Technical Success 

Overall, technical success was achieved in 98.5% of patients. EUS-GE and ES failed in one (1.5%) and two (1.5%) patients, respectively (*p* = 1). 

#### 3.1.3. Clinical Efficacy 

Five patients (two in the EUS-GE group and three in the ES group) died during their hospital stay due to complications that were deemed to be not related to the procedure. In these patients, refeeding was not attempted and they were excluded. Therefore, clinical efficacy was analyzed in 190 patients. Overall, clinical success was achieved in 161 patients (84.7%): 60 (95.2%) in the EUS-GE group and 101 (79.5%) in the ES group (*p* = 0.0006). As reported in Table 4, in the multivariate analysis, PS-ECOG ≥ 2 was an independent factor associated with clinical failure (OR 2.87 [95%CI, 1.08–7.63; *p* = 0.034]).

#### 3.1.4. Safety

AEs were observed in 34 (17.4%) patients without any statistically significant difference between the two groups (*p* = 0.640). Procedure-related AEs occurred in five (2.6%) patients: one in the EUS-GE group (1.5%) and four in the ES group (3.1%) (*p* = 0.666). All procedure-related AEs were late, mild (grade I or grade II according to AGREE classification [25]) bleeding not requiring additional endoscopic or radiologic intervention nor transfusions. Post-procedural AEs occurring during their hospital stay and with a non-obvious (or uncertain) relation to the procedure were observed in 29 patients (14.9%): 12 in the EUS-GE group (18.5%) and 17 in the ES group (13.1%). In the EUS-GE group, all were infectious complications, including sepsis from central venous catheter infection (bacterial or mycotic) in five cases, post-ERCP cholecystitis in three, and pneumonia in two. In two patients, cholangitis occurred after percutaneous biliary drainage with a fatal outcome. In the ES group, complications included sepsis due to post-ERCP cholecystitis or cholangitis in three and four cases, respectively, venous catheter infection in four cases, pulmonary disease (pneumonia and embolism) in two cases, and post-ERCP acute pancreatitis in one case. In one patient, after percutaneous biliary drainage, and two patients with advanced disease (carcinosis and cachexia), the clinical course worsened and death occurred.

#### 3.1.5. Hospital Stay

A significantly shorter hospital stay was observed in the EUS-GE group (7.2 ± 5 days) compared with the ES group (13.9 ± 14.7) (*p* = 0.0005). 

Table 5 summarizes the study outcomes of the main population.

### 3.2. Matched Populations

After matching for age, sex, PS-ECOG, pre-procedural GOOSS, type of malignancy, presence of biliary obstruction, and ascites, 45 patients were allocated to each group, without a baseline difference between the two groups (Table 6).

Technical success was 100% for both groups. Stent dysfunction requiring reintervention occurred less frequently in the EUS-GE group compared with the ES group (4.4% vs. 20.0%; *p* = 0.022). No statistically significant difference was observed between EUS-GE and ES for clinical efficacy (95.6% vs. 86.7%; *p* = 0.266) and AEs rate (15.5% vs. 33.3%; *p* = 0.085). 

A significantly longer hospitalization time was observed after the ES procedure compared with EUS-GE (*p* = 0.018). Outcomes of the matched populations are reported in Table 7.

The stent dysfunction-free survival rate by Kaplan–Meier analysis is shown in Figure 5. Kaplan–Meier analysis showed that EUS-GE experienced a longer time to stent dysfunction compared with ES (log-rank test; *p* = 0.05). The median LAMS and SEMS patency were 69 days (IQR 30–166.5) and 82 days (IQR 41.5–173), respectively. The median survival after EUS-GE was 75 days (30–186.5) vs. 108 days (IQR 49–300) after ES. Kaplan–Meier analysis showed a higher probability of dysfunction-free survival at 3 and 6 months for EUS-GE 96.1% (75.7–99.4) and 88.7% (59.7–97.3) compared with ES 83.1% (65.3%–92.3%) and 78.7% (59.5–89.6%), respectively.

## 4. Discussion

In this study, we compared two endoscopic procedures for the management of mGOO. We collected data from a large cohort of consecutive patients who underwent EUS-GE or ES at two referral Italian institutions. All EUS-GE were performed using 20 mm diameter LAMS, which has been associated with better dietary tolerance compared with 15 mm diameter LAMS [28]. Moreover, the direct free-hand technique, which does not require any specific device [5], was used by endoscopists at both institutions. The study population was analyzed using logistic regression and a propensity score-matched analysis was performed to reduce the risk of selection bias. The primary outcome was the rate of stent dysfunction. This endpoint is clinically relevant because stent dysfunction requires endoscopic or surgical reintervention, a need for further hospitalization, and chemotherapy interruption. Additionally, reintervention adds to the overall cost and economic burden. 

In the present study, considering the whole study population, we found a significantly higher rate of stent dysfunction in the ES group compared with the EUS-GE group (16.9 vs. 3.1%, respectively). This result was confirmed in our multivariate analysis where ES was an independent factor associated with stent dysfunction (OR 8.549 [95% CI, 1.882–38.822]). Similar results were obtained in the matched population, with an even higher difference between ES (stent dysfunction rate of 20%) and EUS-GE (stent dysfunction rate of 4.4%). 

Our findings are in line with the published literature. A propensity score-matching study by van Wanrooij et al. [19] reported 1% of recurrent GOO after EUS-GE compared with 26% after ES. Moreover, in another propensity score-matched population comparing EUS-GE and ES, Vanella et al. [20] reported a symptom recurrence rate significantly lower in the EUS-GE group (3.7% vs. 33.3%, respectively). Finally, in a recently published randomized trial, the reintervention rate due to stent dysfunction was 4% in the EUS-GE group and 29% in the duodenal stent group [22]. Notably, in this randomized trial, an intention-to-treat analysis was performed and reintervention in the EUS-GE group was needed in two patients who ultimately received a duodenal stent due to technical failure from the EUS procedure. However, the follow-up time was 6 months; thus, long-term stent dysfunction rates are unknown. 

All the above-mentioned studies employed an uncovered SEMS for ES. To the best of our knowledge, there are no comparative studies between EUS-GE and covered or partially covered duodenal SEMS. Covered SEMS could theoretically reduce the rate of stent occlusion due to tumor ingrowth. However, two RCTs demonstrated no difference between covered [29] and partially covered [30] duodenal stents compared with uncovered ones.

Additionally, in the present study, we found that a stenosis located in the third part of the duodenum independently increased the risk of stent occlusion. The impact of the site of stenosis on duodenal stent dysfunction was described in previous studies [31,32]. A distal stenosis can be more challenging to deal with during ES and could also reduce the chance of technical success of EUS-GE by impairing the Treitz region that is frequently targeted during the procedure. However, in this setting, EUS-GE should be tried and preferred over ES because of the possibility of finding a distant loop to be punctured and because of the high risk of enteral stent occlusion.

We also evaluated the clinical efficacy of these procedures. When considering the whole study population, we found that EUS-GE was more effective in resolving symptoms and allowing refeeding compared with ES. Biliary obstruction seemed to negatively impact on clinical efficacy in univariate analysis. It is plausible that its occurrence and, consequently, procedures carried out to palliate jaundice/cholangitis may result in a persistent lack of appetite, nausea, and a delayed resumption of solid food intake. Additionally, multivariate analysis showed that clinical success was significantly impacted by PS-ECOG. PS-ECOG represents one of the most important factors evaluated to choose a chemotherapy regimen and could be related to gastroenteric functionality, including peristalsis. The association between performance status and clinical efficacy of ES is well known from previous studies [33,34]. The results of our study suggest that PS-ECOG should be taken into account before proceeding with EUS-GE because of the risk of clinical failure. 

Interestingly, we found that clinical efficacy was similar in the matched population after balancing PS-ECOG between the two groups. This point may also be the reason to explain the contrasting results between our study and the previous literature in terms of clinical success. In fact, in van Wanrooij et al. [19], the clinical success rate was higher after EUS-GE (91%) than after ES (75%). Similarly, in another propensity-matched study [20], Vanella et al. reported 100% clinical success after EUS-GE versus 75% after ES. However, neither van Wanrooij nor Vanella considered PS-ECOG as a clinically relevant variable for propensity matching but they used the ASA score as a scale for the evaluation of patients’ general conditions, and this difference may explain the disagreement with the present study. In contrast, in the first randomized trial published on this topic where only patients with a PS-ECOG of 0–3 were included, no difference in clinical success (defined as an improvement of at least one point in GOOSS within 3 days after stent insertion) was observed (100% after EUS-GE and 92% after ES; *p* = 0.117). Therefore, in very compromised patients with a high PS-ECOG and a life expectancy of less than 3 months, it is reasonable to employ ES over EUS-GE [17].

Safety is one of the major concerns when EUS-GE is performed. Indeed, LAMS misdeployment can lead to severe adverse events requiring surgical intervention [16,21]. In the present study, we divided AEs according to their relationship with the procedure. Overall, a higher rate of AEs was observed in the ES group compared with EUS-GE (20% vs. 16%, respectively), although a statistically significant difference was not reached. Interestingly, considering procedure-related AEs, we did not observe any LAMS misdeployment but only one case of mild bleeding in the EUS-GE group. However, this can be explained by the experience of endoscopists, and the possibility of severe AEs should always be considered when performing EUS-GE. Indeed, misdeployment requiring surgical intervention was reported in 5.7% and 3.4% of cases by Vanella et al. [20] and van Wanrooij et al. [19], respectively. In contrast, no LAMS misdeployment was reported in the RCT by Teoh et al. [22]. However, in this RCT, EUS-GE was performed using a specific device for double-balloon-occluded gastrojejunostomy bypass which is not available worldwide and appears to facilitate LAMS placement. Finally, in our study, four cases of procedure-related bleeding were observed in the ES group vs. one in the EUS-GE group, without a significant difference, despite considering neither the whole nor the matched population. In all cases, AEs were managed conservatively.

Furthermore, our study showed that EUS-GE was associated with significantly less hospital stay compared with ES (7 days vs. 14 days, respectively). Contrasting results have been published regarding this endpoint. In the propensity score-matched study by van Wanrooij et al. [19], a comparable length of hospital stay was observed (4 days after EUS-GE vs. 4 days after ES), whereas Teoh et al. [21] reported a shorter hospitalization time after EUS-GE (median of 4 days) compared with the ES group (median of 6 days). In the present study, the shorter hospital stay in the EUS-GE group was attributable to the lower rate of AEs unrelated to the procedure, as well as better dietary tolerance. 

Enteral stenting can be susceptible to stent dysfunction, stent migration, and stent occlusion, which may necessitate stent revision and even readmission to the hospital for a repeated procedure. However, our study and other published reports indicate that EUS-GE is linked to fewer stent complications (such as those mentioned above) and a reduced need for reintervention. While EUS-GE may be more technically intricate and potentially more expensive, the capacity to mitigate these risks could result in substantial cost savings in the long run [35,36]. Cost-effectiveness modelling studies are needed.

There are some limitations of this study. First, there is an intrinsic selection bias risk due to the retrospective nature of the study without randomization. Indeed, there were significant imbalances of baseline variables in the main population. To mitigate this bias, we performed a multivariate analysis including the whole population and a 1:1 match was created using propensity score analysis. The propensity score represents the probability of each individual patient being assigned to a particular condition in a study given a set of known covariates. With this regard, we included numerous covariates that are known to impact clinical outcomes despite some variables, such as the stage of the disease, not being used. Second, the retrospective retrieval of the same variables, including PS-ECOG, may be difficult in retrospective evaluations, although it was standardly reported as part of the clinical evaluation at the involved institutions. Third, EUS-GE were performed by expert endoscopists at two referral centers, and it is uncertain if such outcomes could be produced by those who are less experienced. Fourth, both EUS-GE and ES techniques are not standardized, especially concerning the site of limb puncture for EUS-GE and the length of SEMS for ES. However, this is common in real world clinical practice where technical aspects are often established on an individual basis. 

## 5. Conclusions

Our study showed that EUS-GE had a lower need for reintervention due to stent dysfunction compared with ES. Moreover, the 3- and 6-month probability of stent patency was higher in the EUS-GE group, suggesting that this procedure might be preferred as first-line in patients with a life expectancy greater than 3 months, especially in those with distal stenosis. On the other hand, ES remains a safe and effective procedure, is widely available, and is easily performed. Overall, the choice between EUS-GE and ES should be based on an individual basis considering several factors associated with the risk of stent dysfunction, clinical success, and local expertise.

## Figures and Tables

**Figure 1 cancers-16-00724-f001:**
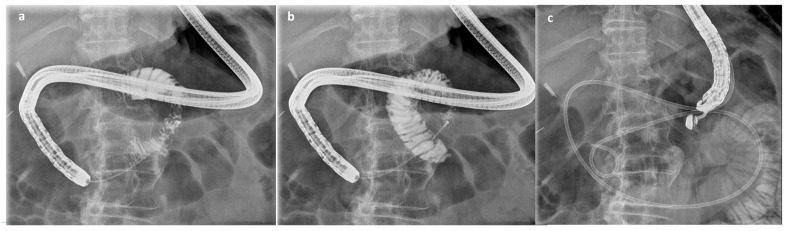
Fluoroscopic steps of EUS-GE procedure: after placing a guidewire and oroenteric catheter, medium contrast is injected into the jejunal loop (**a**,**b**) and nasobiliary drainage is placed over the stenosis (**c**).

**Figure 2 cancers-16-00724-f002:**
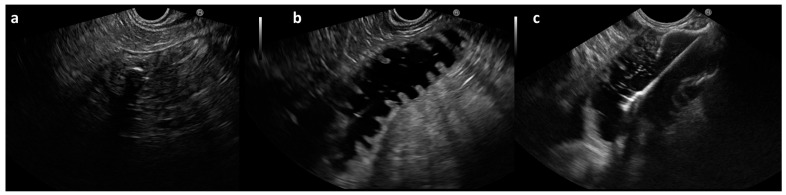
EUS-guided steps of EUS-GE procedure: After identification of the target limb, usually with visible nasobiliary drainage inside (**a**), the lumen is irrigated and distended with methylene blue and saline. (**b**) A lumen apposing metal stent is used to create the anastomosis (**c**).

**Figure 3 cancers-16-00724-f003:**
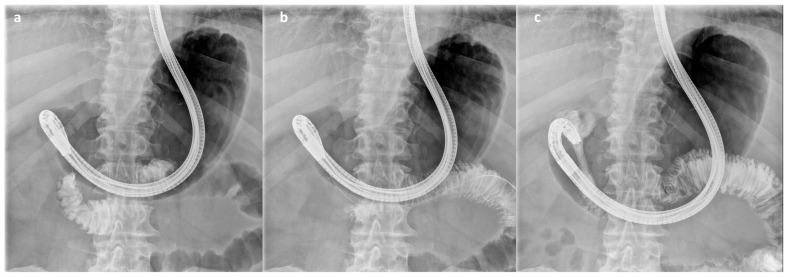
Steps of the ES procedure: After placing a catheter over a guidewire, a contrast agent is injected to estimate the length of the stenosis (**a**). After repositioning the guidewire (**b**), a nitinol uncovered metal stent is deployed and the correct placement is checked with an injection of contrast medium over the stent (**c**).

**Figure 4 cancers-16-00724-f004:**
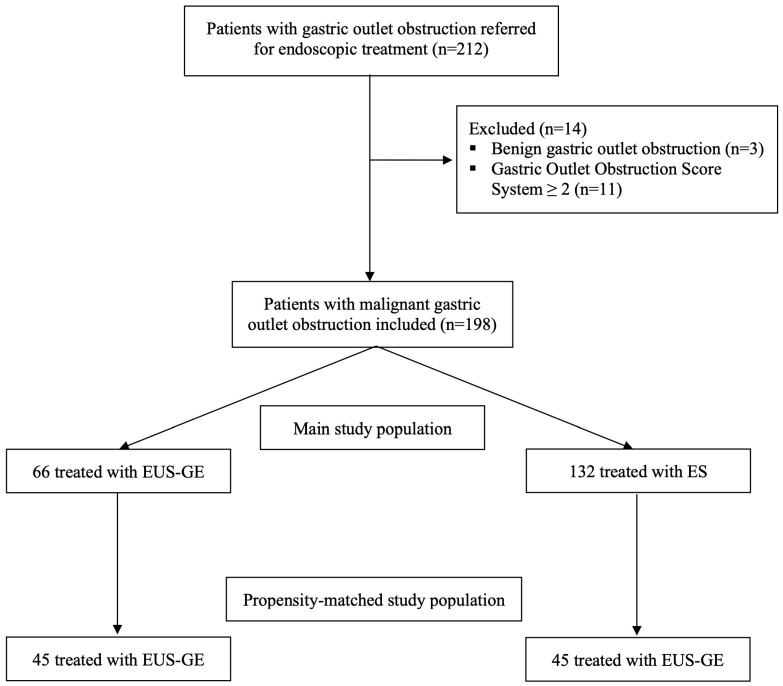
Study flow-chart. EUS-GE: endoscopic ultrasound-guided gastroenterostomy; ES: enteral stenting.

**Figure 5 cancers-16-00724-f005:**
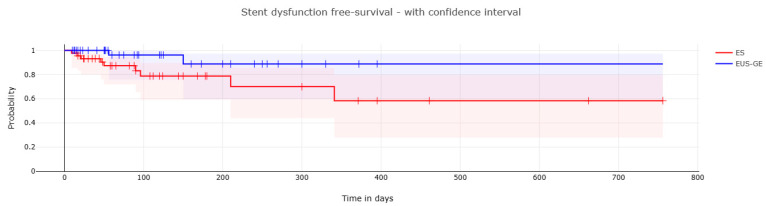
Kaplan–Meier estimates of stent dysfunction-free survival in patients with malignant gastric outlet obstruction who underwent endoscopic ultrasound-guided gastroenterostomy (EUS-GE) or enteral stenting (ES).

**Table 1 cancers-16-00724-t001:** Gastric Outlet Obstruction Score System.

Level of Oral Intake	Score
No oral intake	0
Liquids only	1
Soft solids	2
Low residue or full diet	3

**Table 2 cancers-16-00724-t002:** Baseline characteristics of 198 patients who underwent endoscopic ultrasound-guided gastroenterostomy and enteral stenting.

Variables	Overall n = 198	EUS-GE n = 66	ES n = 132	*p* Value °
Age (years), mean ± SD	67.2 ± 11.4	69.6 ± 11.4	66 ± 11.9	0.045
Sex Male Female	96 (48.5%) 102 (51.5%)	38 (57.6%) 28 (42.4%)	68 (51.5%) 64 (48.5%)	0.420
PS-ECOG 0–1 2 3–4	91 (46.0%) 64 (32.3%) 43 (21.7%)	41 (62.1%) 16 (24.2%) 9 (13.7%)	50 (37.9%) 48 (36.4%) 34 (25.7%)	0.005
Pre-procedural GOOSS * 0 1 Mean value ± SD	132 (66.7%) 66 (27.7%) 0.38 ± 0.59	31 (47.0%) 35 (50.0%) 0.56 ± 0.55	101 (76.5%) 31 (16.7%) 0.25 ± 0.5	<0.0001
Type of malignancy Pancreatic Other ^#^	139 (70.2%) 59 (29.8%)	38 (57.6%) 28 (42.4%)	101 (76.5%) 31 (23.4%)	0.01
Stage of disease Resectable/locally advanced Metastatic	100 (50.5%) 98 (49.5%)	32 (48.5%) 34 (51.5%)	68 (51.5%) 64 (48.5%)	0.373
Ascites Yes No	34 (17.2%) 164 (82.8%)	2 (3.0%) 64 (97.0%)	32 (24.2%) 100 (75.8%)	<0.001
Biliary obstruction Yes No	128 (64.6%) 70 (35.4%)	32 (48.5%) 34 (51.5%)	96 (72.7%) 36 (27.3%)	<0.001
Stenosis location ** Type I Type II Type III	70 (35.4%) 82 (41.4%) 46 (23.2%)	20 (30.3%) 26 (39.4%) 20 (30.3%)	50 (37.9%) 56 (42.4%) 26 (19.7%)	0.229
Follow-up duration, median (IQR), days	90.5 (37–247)	91.5 (42–189)	92 (34–252)	0.894

EUS-GE: endoscopic ultrasound-guided gastroenterostomy; ES: enteral stenting; SD: standard deviation; PS-ECOG: Eastern Cooperative Oncology Group performance status; GOOSS: gastric outlet obstruction score system. ° ES vs. EUS-GE. * According to Adler and Baron [23]; ^#^ Other includes duodenal (18, 9%), biliary (14, 7.1%), gastric (13, 6.6%), papillary (7, 3.5%), metastases (5, 2.6%), retroperitoneal tumors (2, 1.0%). ** According to the Mutignani classification [26].

**Table 3 cancers-16-00724-t003:** Univariate and multivariate analyses of factors impacting stent dysfunction in 195 patients who underwent successful endoscopic ultrasound-guided gastroenterostomy or enteral stenting.

Variables	Univariate Analysis		Multivariate Analysis	
	Stent Dysfunction Yes/No	*p* Value	*p* Value	OR (95% CI)
Age <65 >65	14/79 10/92	0.265	-	
Gender Male Female	14/79 10/92	0.265	-	
Type of malignancy Pancreatic Other	20/121 4/50	0.232	-	
Stage of disease Resectable/locally advanced Metastatic	14/86 10/85	0.460	-	
Ascites Yes No	2/32 22/139	0.263	-	
Biliary obstruction Yes No	18/106 6/65	0.261	-	
Stenosis location * Type I Type II Type III	9/60 5/75 10/36	0.037	0.009	3.49 (1.35–9.02)
PS-ECOG 0–1 2–4	9/79 15/92	0.422	-	
Endoscopic treatment EUS-GE ES	2/63 22/108	0.004	0.005	8.55 (1.88–38.82)

EUS-GE: endoscopic ultrasound-guided gastroenterostomy; ES: enteral stenting; SD: standard deviation; PS-ECOG: Eastern Cooperative Oncology Group performance status. * According to the Mutignani classification [26].

**Table 4 cancers-16-00724-t004:** Univariate and multivariate analyses of factors associated with clinical efficacy in 190 patients who attempted refeeding after successful endoscopic ultrasound-guided gastroenterostomy or enteral stenting.

Variables	Univariate Analysis		Multivariate Analysis	
	Clinical Efficacy Yes/No	*p* Value	*p* Value	OR (95% CI)
Age <65 >65	62/13 99/16	0.541	-	
Gender Male Female	73/17 88/12	0.227	-	
Type of malignancy Pancreatic Other	117/22 44/7	0.822	-	
Stage of disease Resectable/locally advanced Metastatic	86/12 75/17	0.460	-	
Biliary obstruction Yes No	96/25 65/4	0.006	0.052	3.07 (0.99–9.53)
Stenosis location * Type I Type II Type III	55/11 66/13 40/5	0.674	-	
PS ECOG 0–1 2–4	80/6 81/23	0.004	0.034	2.87 (1.08–7.63)
Endoscopic treatment EUS-GE ES	60/3 101/26	0.004	0.056	3.44 (0.96–12.25)

EUS-GE: endoscopic ultrasound-guided gastroenterostomy; ES: enteral stenting; SD: standard deviation; PS-ECOG: Eastern Cooperative Oncology Group performance status. * According to the Mutignani classification [26].

**Table 5 cancers-16-00724-t005:** Outcomes of 195 patients who underwent successful endoscopic treatment for malignant gastric outlet obstruction with endoscopic ultrasound-guided gastroenterostomy or enteral stenting.

Outcomes	Overall n = 195	EUS-GE n = 65	ES n = 130	*p* Value °
Stent dysfunction	24 (12.3%)	2 (3.1%)	22 (16.9%)	0.004
Clinical efficacy	166 (85.1%)	63 (96.9%)	103 (79.2%)	0.0006
Adverse events Overall Procedure-related Not procedure-related	34 (17.4%) 5 (2.6%) 29 (14.9%)	13 (20.0%) 1 (1.5%) 12 (18.5%)	21 (16.2%) 4 (3.1%) 17 (13.1%)	0.640 0.666 0.393
Hospital stay Mean time (days) ± SD	11.8 ± 12.7	7.2 ± 5.0	13.9 ± 14.7	0.0005

° ES vs. EUS-GE. EUS-GE: endoscopic ultrasound-guided gastroenterostomy; ES: enteral stenting; SD: standard deviation.

**Table 6 cancers-16-00724-t006:** Baseline features of the population of patients after propensity score-matching (n = 90) who underwent endoscopic ultrasound-guided gastroenterostomy (n = 45) or enteral stenting (n = 45).

Variables	EUS-GE n = 45	ES n = 45	*p* Value °
Age (years), mean ± SD	68.9 ± 11.5	70.0 ± 10.0	0.523
Sex Male Female	18 (40.0%) 27 (60.0%)	21 (46.7%) 24 (53.3%)	0.670
PS-ECOG 0–1 2 3–4	25 (55.6%) 13 (28.9%) 7 (15.6%)	23 (51.1%) 19 (42.2%) 3 (6.7%)	0.281
Charlson Comorbidity Index ≤7 >7	27 (60.0%) 18 (40.0%)	30 (66.7%) 15 (33.3%)	0.662
Pre-procedural GOOSS * 0 1	18 (40.0%) 27 (60.0%)	25 (55.6%) 20 (44.4%)	0.205
Type of malignancy Pancreatic Other	30 (66.7%) 15 (33.3%)	37 (82.2%) 8 (17.8%)	0.147
Stage of disease Resectable/locally advanced Metastatic	22 (48.9%) 23 (51.1%)	24 (53.3%) 21 (46.7%)	0.123
Ascites No Yes	42 (93.3%) 3 (6.7%)	44 (97.8%) 1 (2.2%)	0.616
Biliary obstruction No Yes	19 (42.2%) 26 (57.8)	22 (48.9%) 23 (51.1%)	0.672
Stenosis location ** Type I Type II Type III	12 (26.7%) 19 (42.2%) 14 (31.1%)	13 (28.9%) 20 (44.4%) 12 (26.7%)	0.935
Follow-up duration, median (IQR), days	80 (52–174)	99 (41.5–192)	0.316

EUS-GE: endoscopic ultrasound-guided gastroenterostomy; ES: enteral stenting; SD: standard deviation; PS-ECOG: Eastern Cooperative Oncology Group performance status; GOOSS: gastric outlet obstruction score system. ° ES vs. EUS-GE. * According to Adler and Baron [23]; ** According to the Mutignani classification [26].

**Table 7 cancers-16-00724-t007:** Outcomes of the population of patients after propensity score-matching (n = 90) who underwent endoscopic treatment for malignant gastric outlet obstruction with endoscopic ultrasound-guided gastroenterostomy or enteral stenting.

Outcomes	EUS-GE n = 45	ES n = 45	*p* Value °
Technical success	45 (100%)	45 (100%)	1
Stent dysfunction	2 (4.4%)	9 (20.0%)	0.022
Clinical efficacy	43 (95.6%)	39 (86.7%)	0.266
Adverse events Overall Procedure-related Not procedure-related	7 (15.5%) 0 (0%) 7 (15.5%)	15 (33.3%) 4 (8.8%) 11 (24.4%)	0.085 0.116 0.429
Hospital stay Mean time (days) ± SD	7.5 ± 4.9	12.5 ± 13.0	0.018

° ES vs. EUS-GE. EUS-GE: endoscopic ultrasound-guided gastroenterostomy; ES: enteral stenting; SD: standard deviation.

## Data Availability

The original contributions presented in the study are included in the article, further inquiries can be directed to the corresponding author.

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
