# Peer review of "Endoscopic Ultrasound-Guided Gastroenterostomy versus Enteral Stenting for Malignant Gastric Outlet Obstruction: A Retrospective Propensity Score-Matched Study"

_cancers, 2024, doi:10.3390/cancers16040724_

Round 1
Reviewer 1 Report
Comments and Suggestions for Authors
Dear Authors and Editors,
thank you for the possibility to revise this Manuscript.
EUS-GE is increasingly demonstrating its advantages over other modalities in the treatment of GOO. This series adds to the compelling evidence (nowadays event prospective and randomized) of reduced dysfunction during follow-up.
The Manuscript is very well written, the inclusion criteria very solid, and statistical analysis reliable. However, I have found some issues that should be corrected/clarified before the acceptance of this paper.
Here attached my comments:
MAJOR
- “Observational study” is not enough to determine the design of a study. The retrospective nature is secondarily stated in the Text. The authors should report the retrospective design in the Title and in the Abstract as well.
- One of the main strengths of this study is the use of prospensity-score matching. Differently to simple matching based on observed differences at univariate analysis, PS matching usually includes variables known to affect the outcome. However, I don’t know a reason why “sex” should affect the dysfunction rate. Moreover I don’t see primary neoplasm (e.g. pancreatic versus gastric) or disease stage (despite the inclusion of ascites, but not carcinosis !) amongst the variables, which might potentially have a stronger impact on outcomes. Indeed, the rate of pancreatic cancers after matching tends to be higher in the ES versus EUS-GE group. Furthermore, the Authors have observed different preprocedural GOOSS in the two arms, and should have corrected for this. Finally, the Authors report as a major strength of their study the use of PS-ECOG as a matching variable. However, this variable is difficult to be retrieved in large-scale retrospective evaluations, unless it is standardly reported as part of the standard clinical evaluation in the Institutions, which is not quite common for endoscopy units. Please clarify, and discuss this limitation in the Discussion.
- Based on their retrospective study, the Authors speculate that the use of PS-ECOG score for matching is the most probable explanation why they found no different clinical success compared to previous literature showing higher clinical success of EUS-GE, claiming that older studies have not considered performance status. However, Vanella et al. has 1:1 matched for prospectively evaluated ASA score. Despite different from the PS-ECOG, the two scores similarly assess with a 5-category score the domain of health status and functional ability. Moreover, this was addressed prospective conversely to the PS-ECOG score in this study. I would therefore soften the conclusions reached in the Discussion.
- I find it surprising that the rate of concomitant biliary obstruction is higher in the ES group. I would have expected a higher use of EUS-GE to avoid conflicting with SEMS. Can the Authors clarify what criteria they’ve used for the allocation in the two treatment arms?
- This is a retrospective study with no scheduled follow-up intervals. We need to know the median FU of the two cohorts, with comparison.
- The Authors report 2 cases of dysfunction, but then say “the two stent dysfunction cases (due to LAMS occlusion) were treated with a coaxial enteral SEMS while the other case was surgical treated for delayed perforation close to the LAMS, respectively”. Please clarify
- “A mean time of 115.6 days from endoscopic procedure to re-intervention was observed (EUS-GE group 103 days and ES group 116.8 days)”. Please report confidence interval and the results of the Mann-Whiney test for this comparison.
- The Authors report the results of a multivariable analysis on the risk of dysfunction, despite they have not described this analysis in the Statistics section (is this a logistic regression? With which specification?). Moreover, I don’t understand the need to perform a multivariable analysis with the primary outcome as the dependent variable and the 2 arms of the study as independent variables. Probably it might have helped in correcting the Kaplan Meier estimation of patency for potentially interfering variables (such as stenosis location) but this has not been performed.
- Do the Authors have an explanation why having biliary obstruction is associated with better alimentary clinical success? Sounds not logical.
- When describing a novel procedure, association of AEs with the procedure should be very cautious and dubitative. I would like the Authors to better describe infectious complications in their cohort and how they assessed that they were not procedure-related, especially as some events were fatal.
- When reporting results of a Kaplan Meier usually sentences such “revealed a significantly higher stent dysfunction rate” is not appropriate. Kaplan Meier evaluates time to events (survival) and therefore I would report differently the results of a positive log-rank.
MINOR
- When dealing with surgical GE I would suggest to change “high rate of clinical success” into “high rate of technical success”, as some of surgical patients are affected by postsurgical delayed gastric emptying.
- Please amend “Less Invasiveness”
- In the ESGE guidelines there is no reference to life expectancy when evaluating EUS-GE versus alternatives. Please amend. “ESGE recommends EUS-guided gastroenterostomy (EUS-GE), in an expert setting, for malignant gastric outlet obstruction, as an alternative to enteral stenting or surgery”.
- Usually “exact matching” refers to standard matching (one male with one male, one pancreatic cancer with one pancreatic cancer) and not propensity-score matching.
Author Response
MAJOR
- “Observational study” is not enough to determine the design of a study. The retrospective nature is secondarily stated in the Text. The authors should report the retrospective design in the Title and in the Abstract as well.
RE: We stated the study's retrospective design in the title, abstract, and methods.
- One of the main strengths of this study is the use of prospensity-score matching. Differently to simple matching based on observed differences at univariate analysis, PS matching usually includes variables known to affect the outcome. However, I don’t know a reason why “sex” should affect the dysfunction rate. Moreover I don’t see primary neoplasm (e.g. pancreatic versus gastric) or disease stage (despite the inclusion of ascites, but not carcinosis !) amongst the variables, which might potentially have a stronger impact on outcomes. Indeed, the rate of pancreatic cancers after matching tends to be higher in the ES versus EUS-GE group. Furthermore, the Authors have observed different preprocedural GOOSS in the two arms, and should have corrected for this. Finally, the Authors report as a major strength of their study the use of PS-ECOG as a matching variable. However, this variable is difficult to be retrieved in large-scale retrospective evaluations, unless it is standardly reported as part of the standard clinical evaluation in the Institutions, which is not quite common for endoscopy units. Please clarify, and discuss this limitation in the Discussion.
RE: Thank you for your comment. We included sex as a variable to have demographically balanced groups. We also apologize for the typo we made. Indeed, the pre-procedural GOOSS and type of malignancy were included as variables for propensity matching, but we forgot to write them in the methods (differently, we correctly reported them in the results section, lines 354/5). We amended the text of the Methods section. About the stage of the disease, we agree that it could impact clinical outcomes, but it was already well-balanced before matching (and it remained well-balanced after matching). However, we agree with your point and added it as a limitation in the discussion. The PS-ECOG is routinely reported in patients’ charts and is easily retrievable in our Institutes. However, due to the retrospective design of the study, we included this point as a limitation in the discussion, as suggested.
- Based on their retrospective study, the Authors speculate that the use of PS-ECOG score for matching is the most probable explanation why they found no different clinical success compared to previous literature showing higher clinical success of EUS-GE, claiming that older studies have not considered performance status. However, Vanella et al. has 1:1 matched for prospectively evaluated ASA score. Despite different from the PS-ECOG, the two scores similarly assess with a 5-category score the domain of health status and functional ability. Moreover, this was addressed prospective conversely to the PS-ECOG score in this study. I would therefore soften the conclusions reached in the Discussion.
RE: We agree with your comment. We soften the conclusions and the discussion as suggested.
- I find it surprising that the rate of concomitant biliary obstruction is higher in the ES group. I would have expected a higher use of EUS-GE to avoid conflicting with SEMS. Can the Authors clarify what criteria they’ve used for the allocation in the two treatment arms?
RE: Thank you for your comment. The rate of biliary obstruction was higher in the main population, whereas it was well-balanced after propensity (as it was one of the variables used). The reason for a higher rate of biliary obstruction in the ES group could be that some patients underwent ES to subsequently reach the papilla and perform ERCP. Second, the study timeframe of enteral stenting was longer (as it is an “older” procedure); thus, a higher rate of biliary obstruction was probably managed with this procedure.
- This is a retrospective study with no scheduled follow-up intervals. We need to know the median FU of the two cohorts, with comparison.
RE: We added the median FU time. Thank you!
- The Authors report 2 cases of dysfunction, but then say “the two stent dysfunction cases (due to LAMS occlusion) were treated with a coaxial enteral SEMS while the other case was surgical treated for delayed perforation close to the LAMS, respectively”. Please clarify
RE: Sorry for the misleading sentence. We amended the text to: “The two stent dysfunction cases were treated with a coaxial enteral SEMS and surgically for concomitant perforation close to the LAMS, respectively.”
- “A mean time of 115.6 days from endoscopic procedure to re-intervention was observed (EUS-GE group 103 days and ES group 116.8 days)”. Please report confidence interval and the results of the Mann-Whiney test for this comparison.
RE: We understand your point. We did not perform a comparison because only 2 patients in the EUS-GE group underwent re-intervention. We removed this sentence because similar information was reported using the Kaplan-Meier that showed a higher probability of dysfunction-free survival at 3 and 6 months.
- The Authors report the results of a multivariable analysis on the risk of dysfunction, despite they have not described this analysis in the Statistics section (is this a logistic regression? With which specification?). Moreover, I don’t understand the need to perform a multivariable analysis with the primary outcome as the dependent variable and the 2 arms of the study as independent variables. Probably it might have helped in correcting the Kaplan Meier estimation of patency for potentially interfering variables (such as stenosis location) but this has not been performed.
RE: In the Methods section, we explained that a “stepwise logistic regression” was used. We performed a multivariate analysis to identify factors associated with stent dysfunction and clinical success in the main population. Therefore, the primary outcome was the dependent variable, and the treatment (EUS-GE or ES) was one of the independent variables. Differently, the Kaplan-Meier analysis was performed to evaluate the clinical outcomes over time.
- Do the Authors have an explanation why having biliary obstruction is associated with better alimentary clinical success? Sounds not logical.
RE: sorry for the lack of clarity. Actually, biliary obstruction was associated with worse clinical success. In fact, as reported in Table 4, 96/121 (79%) patients with biliary obstruction vs 65/69 (94%) without biliary obstruction had clinical success.
- When describing a novel procedure, association of AEs with the procedure should be very cautious and dubitative. I would like the Authors to better describe infectious complications in their cohort and how they assessed that they were not procedure-related, especially as some events were fatal.
RE: Thank you for your comment. We added a precise description of AEs in both the EUS-GE and ES cohorts.
- When reporting results of a Kaplan Meier usually sentences such “revealed a significantly higher stent dysfunction rate” is not appropriate. Kaplan Meier evaluates time to events (survival) and therefore I would report differently the results of a positive log-rank.
RE: Thank you for your comment. We changed the sentence to: “Kaplan–Meier analysis showed that EUS-GE experienced a longer time to stent dysfunction when compared with ES (log-rank test, p= 0.05)”
MINOR
- When dealing with surgical GE I would suggest to change “high rate of clinical success” into “high rate of technical success”, as some of surgical patients are affected by postsurgical delayed gastric emptying.
RE: Done, thank you!
- Please amend “Less Invasiveness”
RE: Done, thank you!
- In the ESGE guidelines there is no reference to life expectancy when evaluating EUS-GE versus alternatives. Please amend. “ESGE recommends EUS-guided gastroenterostomy (EUS-GE), in an expert setting, for malignant gastric outlet obstruction, as an alternative to enteral stenting or surgery”.
RE: Done, thank you!
- Usually “exact matching” refers to standard matching (one male with one male, one pancreatic cancer with one pancreatic cancer) and not propensity-score matching.
RE: Thank you, we removed “exact” from the text.
Reviewer 2 Report
Comments and Suggestions for Authors
This multicenter retrospective Italian cohort study has compared clinical results between EUS-GE and ES. They have found some advantages of EUS-GE, such as patency, stent dysfunction-free survival, and duration of hospital day. It is an exciting study that adds evidence to the field of study. However, there are some concerns about this article. 1. Please present the name of the endoscope that they used. 2. To help the readers understand, please present figures of the stepwise procedure of LAMS and SEMS from their original cases. 3. How many interventional EUS were performed in their facilities during the study periods? What ratio, including the study patients? 4. The authors should unite many tables to clarify the results.
Comments on the Quality of English LanguageMinor English editing is required.
Author Response
RE: thank you for your suggestions. 1. The procedure paragraph was implemented with the endoscopes used. 2. Figures of both EUS-GE and ES were added in the procedure section. (Fig. 1, 2 and 3) 3. Two tertiary centers with high expertise in interventional EUS (over 100 procedures /per year) participated in the study. We specified it in the text. EUS-GE has been introduced in the last 3 years. All 80 procedures performed during the study period were considered for inclusion, but 14 were excluded, as reported in the study flowchart. 4. Thank you for your comment, but we believe it is not advisable to unify the tables, as the different purposes and populations involved. Some percentages were added to make it clearer.
Reviewer 3 Report
Comments and Suggestions for Authors
In the present article Conti Bellocchi et al compared enteral stenting (ES) to EUS-guided gastroenterostomy (EUS-GE) for patients with malignant gastric outlet obstruction. After propensity score matching, they found that EUS-GE was comparable to ES in terms of clinical efficacy and safety, but allowed less stent dysfunction, longer stent patency and shorter hospital stay. Main comments:
1) A linguistic revision is necessary (see for instance page 1 line 22: require - - > requires).
2) I suggest to precise in the Abstract that EUS—GE was performed by placing LAMS.
3) Please add a reference for PS-ECOG.
4) Table 2: “Other” is a quite generic as malignancy etiology. Please give more details.
5) In table 2, please report whether p values refer to the comparison ES vs EUS-GE.
6) Which was the median follow up time in the two patients groups?
7) How could the Authors explain the finding that biliary obstruction negatively impacted on clinical efficacy, as reported in table 4?
8) A short discussion about the costs of ES and EUS-GE should be added.
Comments on the Quality of English LanguageSee above
Author Response
- A linguistic revision is necessary (see for instance page 1 line 22: require - - > requires).
RE: Thank you for your comment. We edited the manuscript.
- I suggest to precise in the Abstract that EUS—GE was performed by placing LAMS.
RE: Done! Thank you
- Please add a reference for PS-ECOG.
RE: Done! Thank you
Table 2: “Other” is a quite generic as malignancy etiology. Please give more details.
RE: Sorry for the lack of clarity. The description of other malignancies was added as a Table footnote.
5) In table 2, please report whether p values refer to the comparison ES vs EUS-GE.
RE: we added a footnote to specify that p values refer to ES vs EUS-GE.
6) Which was the median follow up time in the two patients groups?
RE: Thank you for your comment. We added the median follow-up time.
7) How could the Authors explain the finding that biliary obstruction negatively impacted on clinical efficacy, as reported in table 4?
RE: Thank you for your question. It is plausible that biliary obstruction and, consequently, procedures carried out to palliate jaundice/cholangitis may result in a persistent lack of appetite, nausea, and a delayed resumption of solid food intake.
8) A short discussion about the costs of ES and EUS-GE should be added.
RE: We added the following paragraph in the discussion: “Enteral stenting can be susceptible to stent dysfunction, stent migration, and stent occlusion, which may necessitate stent revision and even readmission to the hospital for a repeated procedure. However, our study and other published reports indicate that EUS-GE is linked to fewer stent complications (such as those mentioned above) and a reduced need for re-intervention. While EUS-GE may be more technically intricate and potentially more expensive, the capacity to mitigate these risks could result in substantial cost savings in the long run. Cost-effectiveness modelling studies are needed.” Thank you!
Round 2
Reviewer 1 Report
Comments and Suggestions for Authors
Dears Editor,
the Authors have done an escellent jor in replying to all comments and I found this version of the Manuscript improved.
I have only one additional suggestion before acceptance. The Authors have now better specified AEs which were not intraprocedural. However, some of them are infections which might be still relatable to the procedure (a post-procedural pneumonia might be an aspiration pneumonia; catheters infection might be due to intraprocedural bacteremia). Therefore I suggest a more prudential label instead of "AEs not related to the procedure". Something like "Post-procedural AEs occurring during hospital stay and with a non-obvious (or uncertain) relation to the procedure ...."
Author Response
Thank you for your suggestion! Done.
Reviewer 2 Report
Comments and Suggestions for Authors
The article has been revised according to the reviewer's comments and is much improved.
Author Response
Thank you.
Reviewer 3 Report
Comments and Suggestions for Authors
Regarding the answer to point 7, the answer should be integrated in the Discussion section.
Regarding the answer to point 8, please add a reference supporting the statement.
All other answers were OK, and the paper may be accepted after these minor changes.
Author Response
Thank you for your suggestions. Done!